# ∞-VIDEO: A Training-Free Approach to Long Video Understanding via Continuous-Time Memory Consolidation

**Saul Santos** [1 2]  **António Farinhas** [1 2]  **Daniel C. McNamee** [3]  **André F. T. Martins** [1 2 4 5]

## Abstract

Current video-language models struggle with long-video understanding due to limited context lengths and reliance on sparse frame subsampling, often leading to information loss. This paper introduces ∞-VIDEO, which can process arbitrarily long videos through a continuous-time long-term memory (LTM) consolidation mechanism. Our framework augments video Q-formers by allowing them to process unbounded video contexts efficiently and without requiring additional training. Through continuous attention, our approach dynamically allocates higher granularity to the most relevant video segments, forming "sticky" memories that evolve over time. Experiments with Video-LLaMA and VideoChat2 demonstrate improved performance in video question-answering tasks, showcasing the potential of continuous-time LTM mechanisms to enable scalable and training-free comprehension of long videos.

## 1. Introduction

Multimodal large language models have driven progress in video-language tasks through the integration of pretrained visual encoders with powerful text-based models (Li et al., 2023b; Zhang et al., 2023b; Cheng et al., 2024; Li et al., 2024). However, current video models are constrained by short context lengths (Li et al., 2023b; Maaz et al., 2024; Liu et al., 2023) and often rely on sparse frame subsampling for longer sequences, which limits their ability to fully process and understand long videos. This contrasts with the high-capacity persistence of human memory (Brady et al., 2008) and the cognitive principles by which humans store information over long timescales, which involve consoli-

dation processes adaptively integrating important episodic events into long-term memory (McGaugh, 2013; Cowan et al., 2021). *How can we ensure models are able to fully understand and grasp information from arbitrarily long videos while they process them without losing critical details?*

Transformers offer significant potential for extracting spatio-temporal features from videos (Zhang et al., 2023b; Li et al., 2024). While recent video-language models have prioritized simpler methods such as projection layers (Li et al., 2023d; Liu et al., 2023; Li et al., 2023b; Liu et al., 2024c; Ye et al., 2024) and temporal pooling (Luo et al., 2023; Maaz et al., 2024) for the sake of efficiency and scalability, these approaches often sacrifice transformer's representational depth. Additionally, training video-language models presents significant difficulties, whereas traditional approaches mainly focus on scaling model parameters (Liu et al., 2024c; Cheng et al., 2024; Li et al., 2024), which requires substantial computational resources. A recent alternative explored the usage of additional computation during inference (Zhang et al., 2023a; Wang et al., 2024a;c). However, these methods assume that the spatio-temporal module is intrinsic to the LLM, which limits their ability to effectively specialize in capturing spatio-temporal content. Furthermore, these methods often rely on subsampling and are designed to process the entire video whenever they need to answer a question.

In this paper, we take an alternative approach inspired by human cognition, where memory consolidation processes enable the retention and efficient handling of long-term dependencies (Frankland & Bontempi, 2005; Preston & Eichenbaum, 2013; Song et al., 2023; 2024; Balazevic et al., 2024). Namely, we develop a new framework with **continuous-time visual memory representations**. Our framework adapts the ∞-former architecture previously developed for textual data (Martins et al., 2020; 2022b), which we leverage to extend the capabilities of pre-trained short-context multimodal LLMs, making them able to process unbounded video contexts in a training-free manner. Shifting from a discrete to a continuous attention framework parallels the recent evolution in theories of human working memory mediated by prefrontal cortex—from the discrete "slot-based" model to the continuous "shared resource" approach (Ma et al., 2014). Consequently, our method aims to cultivate

[1]Instituto de Telecomunicações [2]Instituto Superior Técnico, Universidade de Lisboa [3]Champalimaud Research [4]ELLIS Unit Lisbon [5]Unbabel. Correspondence to: Saul Santos <saul.r.santos@tecnico.ulisboa.pt>.

*Proceedings of the 42$^{nd}$ International Conference on Machine Learning*, Vancouver, Canada. PMLR 267, 2025. Copyright 2025 by the author(s).

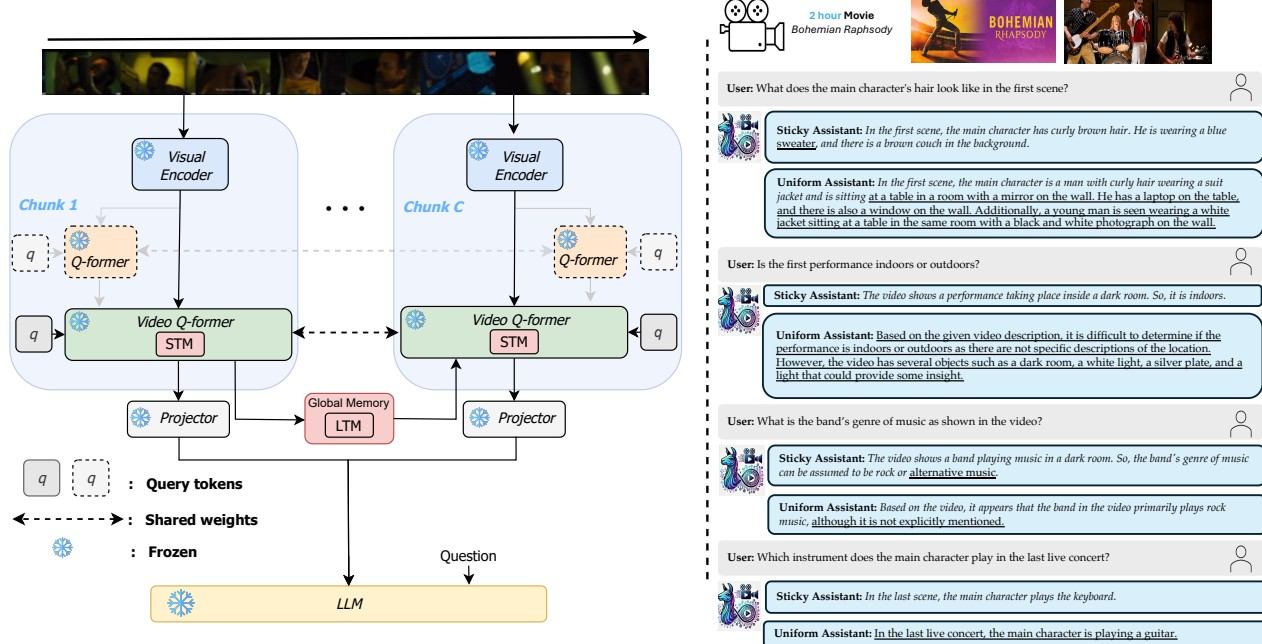

*Figure 1.* **(Left)** Overview of ∞-VIDEO (our approach) using Video LLaMA (Zhang et al., 2023b, gray arrows), which uses an additional spatial Q-former module, and VideoChat2 (Li et al., 2024, black arrows). We split the video into frame chunks and apply these models to each chunk. The Video Q-former module combines a weighted average of the STM, which is the attention for an individual chunk, with a continuous LTM that takes into account previous chunks. The outputs of the Video Q-Former are projected and then averaged. The LLM takes as input visual tokens, generated by our modified video Q-former, alongside the corresponding question to obtain the answer. **(Right)** Examples of ∞-Video LLaMA answers, equipped with our LTM, with uniform sampling and sticky memories for short and ultra-long videos. *Italicized* corresponds to the correct answer, while underlined corresponds to a wrong answer or hallucination.

similar insights within the realm of episodic memory processing, where dynamic handling of memory is essential. This leads to ∞-VIDEO models (Fig. 1), which are able to process and organize information as it comes, with only one pass over the video. Our main contributions are:[1]

- We equip the current attention mechanism of video Q-formers (short-term memory, STM) with a continuous-time LTM that consolidates video information by dynamically allocating higher granularity to the most relevant parts of a video.

- We develop a new continuous-time attention mechanism which is more powerful than the Gaussian model of Martins et al. (2022b) by considering the Gibbs density based on the continuous query-key similarity function.

- We show that architectures with spatio-temporal feature extractors designed for short videos can generalize to long-video understanding in a simple, training-free manner without requiring task-specific fine-tuning or training on

long-video datasets.

- We validate our approach using the Video-LLaMA (Zhang et al., 2023b) and VideoChat2 (Li et al., 2024) models by processing a stream of video frames with a single pass, where the STM considers one chunk at the time, and the LTM maintains global information from past chunks. Our proposed model shows improved performance on video question-answering tasks when using the LTM and competitive results with other training-free models.

## 2. Background

### 2.1. Discrete Attention

Attention mechanisms (Bahdanau et al., 2015) act as a memory component in modern neural networks, enabling them to dynamically focus on key parts of the input and capture long-range dependencies, improving performance across various tasks (Vaswani et al., 2017; Dosovitskiy et al., 2021).

Consider two sequences $\boldsymbol{X} \in \mathbb{R}^{L \times e}$ and $\boldsymbol{Y} \in \mathbb{R}^{R \times e}$, where $L$ and $R$ are the sequence lengths and $e$ is the embedding size. A vanilla attention mechanism in a transformer works

---

[1]The code is made available in https://github.com/deep-spin/Infinite-Video.

as follows. First, we obtain queries ($Q$), keys ($K$), and values ($V$) by linearly projecting $X$ and $Y$ for each attention head $h$:

$$Q^h = Y W_Q^h, \quad K^h = X W_K^h, \quad V^h = X W_V^h, \quad (1)$$

where $W_Q^h \in \mathbb{R}^{e \times d}$, $W_K^h \in \mathbb{R}^{e \times d}$, and $W_V^h \in \mathbb{R}^{e \times d}$ are head-specific learnable projection matrices, $d = e/|h|$, and $|h|$ is the number of attention heads. For each head, the context representation $Z^h \in \mathbb{R}^{L \times d}$ is computed as:

$$Z^h = \text{softmax}\left(\frac{Q^h (K^h)^\top}{\sqrt{d}}\right) V^h. \quad (2)$$

The outputs from all heads are then concatenated to obtain the final context representation $Z \in \mathbb{R}^{L \times e}$:

$$Z = \begin{bmatrix} Z^1 & Z^2 & \cdots & Z^{|h|} \end{bmatrix} W_Z, \quad (3)$$

where $W_Z \in \mathbb{R}^{e \times e}$ is another learnable projection matrix.

## 2.2. Continuous Attention

Instead of splitting the input object into a finite set of pieces (*e.g.*, tokens in text or pixels in images), continuous attention mechanisms (Martins et al., 2020) assume an underlying continuous domain, suitable for arbitrarily long temporal signals, such as audio or video data. This is done by replacing the attention probability mass function by a probability *density* function (PDF) over a continuous signal.

In continuous attention, the input is assumed to be a continuous signal $x(t)$. Although video data is "continuous-time" in nature, it comes as a stream of $L$ discrete frames $X = [x_1^\top, ..., x_L^\top] \in \mathbb{R}^{L \times e}$, and therefore it is necessary to convert this sequence into a smooth continuous signal. This can be done by expressing the continuous signal $x(t) \in \mathbb{R}^e$ as a linear combination of $N$ basis functions $\psi(t) \in \mathbb{R}^N$:

$$x(t) = B^\top \psi(t), \quad (4)$$

where $B \in \mathbb{R}^{N \times e}$ is a coefficient matrix. For compression, it is appealing to use a smaller number of basis functions than frames, $N \ll L$. $B$ can be computed with multivariate ridge regression (Brown & Zidek, 1980), where we set the domain of the continuous-time signal to the unit interval $[0, 1]$. Frames are associated with time instants in this unit interval, $t_1 \leq t_2 \leq ... \leq t_L$, with each $t_\ell \in [0, 1]$, and we set our design matrix as $F = [\psi(t_1), \ldots, \psi(t_L)] \in \mathbb{R}^{N \times L}$. The coefficients $B$ are computed such that $x(t_\ell) \approx x_\ell$ for each frame $\ell \in \{1, \ldots, L\}$, with $\lambda > 0$, leading to:

$$B^\top = X^\top F^\top (F F^\top + \lambda I)^{-1}. \quad (5)$$

The final step involves attending to $x(t)$. In this approach, a PDF $p(t)$ replaces the probability mass function of the discrete attention.[2] The context is then computed as the expected value of the values $v(t) = (W_V)^\top x(t)$:

$$Z = \mathbb{E}_p[v(t)]. \quad (6)$$

**∞-former.** In a transformer with discrete attention, handling a long context (large $L$) becomes impractical due to excessive memory demands. The ∞-former (Martins et al., 2022b) overcomes this limitation by means of an unbounded LTM leveraging continuous attention (§2.2). It allows for unbounded context without increasing memory usage by trading off the number of basis functions that fit into memory with the granularity of their representations. It achieves this by sampling points within the $[0, 1]$ interval, either uniformly or based on prior attention, at which $x(t)$ is evaluated. This process, as we will see in §3, can be seen as the memory consolidation step, allowing new information from the short-term memory $x(t)$ to be incorporated by scaling it down with a forgetting factor $\tau$. The past context is associated with positions in $[0, \tau]$, while the new context is associated in $(\tau, 1]$, followed by ridge regression over the new $x(t)$ and computation of the output context as in Eq. 6.

## 2.3. Video Q-former

A video Q-former (Zhang et al., 2023b) is a specialized variant of a transformer architecture designed to enable LLMs to process and understand video content. Developed initially to capture spatial features in images (Li et al., 2023a), its primary function is to map $L \times P$ spatial embeddings, where $P$ can be the number of patch vectors from a visual encoder, to $R$ spatio-temporal video representations. It operates by stacking layers that first apply self-attention between $R$ learned queries, followed by cross-attention between this self-attention output and the $L \times P$ spatial embeddings, leading to $R$ spatio-temporal representations.

## 3. Unbounded Memory Video Q-former

In this section, we describe our approach to endow video models with a continuous-time memory mechanism. We adapt existing video Q-former models (Zhang et al., 2023b; Li et al., 2024) by first splitting the full sequence of frames into chunks and then processing each chunk individually. Each chunk includes a discrete STM, which is the already present cross-attention in that chunk. However, building the LTM assumes continuity within the embeddings, which is not the case since each frame contains $P$ distinct embeddings. To address this, we perform average pooling over the $P$ embeddings, resulting in a discrete sequence $X \in \mathbb{R}^{M \times e}$, where $M$ denotes the number of frames in the chunk. We

---

[2]For instance, in Martins et al. (2022b), the PDF is modeled as a Gaussian distribution $\mathcal{N}(t; \mu, \sigma^2)$, where the parameters $\mu$ and $\sigma^2$ are learned by a neural network. In §3, we propose a different, more powerful strategy.

introduce a global and dynamic LTM, which works with a modified, more powerful version of the continuous attention mechanism in §2.2. The LTM update enables increased granularity in memory regions with higher cumulative attention density.

### 3.1. Long-Term Memory

The first step in building our continuous LTM is to project the long-term continuous input $\boldsymbol{x}(t) = \boldsymbol{B}^\top \boldsymbol{\psi}(t)$, leading to the continuous keys $\boldsymbol{k}^h(t) \in \mathbb{R}^d$ and values $\boldsymbol{v}^h(t) \in \mathbb{R}^d$, with the same projection matrices (1) of the STM. This allows us to perform attention over the same embedding space as the vanilla video Q-former as:

$$\boldsymbol{k}^h(t) = (\boldsymbol{W}_K^h)^\top \boldsymbol{x}(t) = (\boldsymbol{W}_K^h)^\top \boldsymbol{B}^\top \boldsymbol{\psi}(t), \quad (7)$$

$$\boldsymbol{v}^h(t) = (\boldsymbol{W}_V^h)^\top \boldsymbol{x}(t) = (\boldsymbol{W}_V^h)^\top \boldsymbol{B}^\top \boldsymbol{\psi}(t). \quad (8)$$

We compute the query $\boldsymbol{Q}^h = [\boldsymbol{q}_1^\top, ..., \boldsymbol{q}_R^\top] = \boldsymbol{Y}\boldsymbol{W}_Q^h$ as in (1). For each query $\boldsymbol{q}_i \in \mathbb{R}^d$, we compute the continuous query-key similarity $s_i^h(t)$ for each head and query as

$$s_i^h(t) = \boldsymbol{q}_i^\top \boldsymbol{k}(t) = \boldsymbol{q}_i^\top (\boldsymbol{W}_K^h)^\top \boldsymbol{B}^\top \boldsymbol{\psi}(t), \quad (9)$$

and compute a Gibbs PDF as

$$p_i^h(t) = \frac{\exp(s_i^h(t))}{\int \exp(s_i^h(t'))\, dt'}, \quad (10)$$

where the integral is approximated with the trapezoidal rule. Given the value function $\boldsymbol{v}^h(t)$, we compute the attention-specific representation vectors as described in (6):

$$\boldsymbol{Z}_i^h = \mathbb{E}_{p_i^h}[\boldsymbol{v}^h(t)] = (\boldsymbol{W}_V^h)^\top \boldsymbol{B}^\top \int p_i^h(t)\boldsymbol{\psi}(t)\, dt. \quad (11)$$

Finally, we obtain the LTM representation $\boldsymbol{Z}_{\text{LTM}}$ by concatenating the context heads and projecting them as in (3).

### 3.2. Continuous-Time Memory Consolidation

As we continue processing chunks, the LTM is progressively updated, as illustrated in Fig. 2. This is done by first sampling $T$ locations within the interval $[0, 1]$ and then evaluating the continuous signal $\boldsymbol{x}(t)$, or current LTM, at these sampled points. The sampling can be performed either uniformly in $[0, 1]$ or based on prior attendance, as we will explain in detail in §3.3. The next step involves concatenating this LTM context with the new one coming from the current chunk. To do this, we first "contract" the LTM as

$$\boldsymbol{x}'(t) = \boldsymbol{x}(t/\tau) = \boldsymbol{B}^\top \boldsymbol{\psi}(t/\tau). \quad (12)$$

We then compute $\boldsymbol{x}(t)$ at the $T$ locations $0 \le t_1, ..., t_T \le \tau$:

$$\boldsymbol{x}_i = \boldsymbol{B}^\top \boldsymbol{\psi}(t_i/\tau) \quad \forall i \in [T], \quad (13)$$

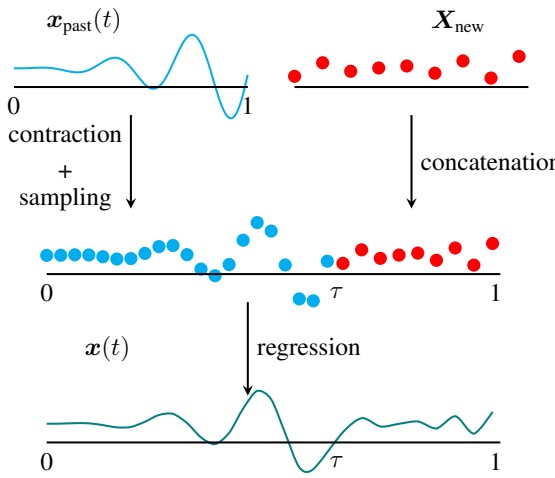

*Figure 2.* Proposed Memory Consolidation Mechanism.

and build the matrix $\boldsymbol{X}_{\text{past}} = [\boldsymbol{x}_1, \boldsymbol{x}_2, ..., \boldsymbol{x}_T]^\top \in \mathbb{R}^{T \times e}$. Following this, we concatenate the current step context $\boldsymbol{X}_{\text{new}} \in \mathbb{R}^{M \times d}$ with the previous context $\boldsymbol{X}_{\text{past}} \in \mathbb{R}^{T \times e}$, resulting in the combined sequence:

$$\boldsymbol{X} = [\boldsymbol{X}_{\text{past}}, \boldsymbol{X}_{\text{new}}]^\top \in \mathbb{R}^{(T+M) \times e}. \quad (14)$$

With the new and previous chunk contexts, we compute $\boldsymbol{B}$, as described in Eq. 5, where the continuous signal is approximated using a linear combination of rectangular functions. To achieve this, we first adjust the contribution of the previous context using the factor $\tau$. Specifically, we associate $\boldsymbol{X}_{\text{past}}$ with positions in the interval $[0, \tau]$ and $\boldsymbol{X}_{\text{new}}$ with positions in $(\tau, 1]$. This process yields a matrix $\boldsymbol{G} \in \mathbb{R}^{(M+L) \times N}$. During each step, the previous context is contracted by a factor of $\tau$, which regulates the extent of long-term memory used for attention and induces a gradual "forgetting" process. After the computation of $\boldsymbol{B}$, continuous attention is performed as described in §3.1.

### 3.3. Sticky Memories

When extending the LTM, the signal is evaluated at $T$ locations within $[0, 1]$, as described in §3.2. While these locations can be uniformly distributed, allocating more "memory space" to regions of higher relevance ensures that critical information is prioritized in line with continuous resource allocation conceptualizations of human memory (Ma et al., 2014). This selective allocation compresses the signal into an efficient representation whereby essential video details are retained, and less significant ones are compressed or discarded. The recurrent and adaptive nature of this process is reminiscent of memory consolidation and reconsolidation brain mechanisms for relevance-based long-term memory transformation (Hardt et al., 2010; Dudai et al., 2015).

To address this, we propose selecting the $T$ locations based

on the relevance of the signal in each region as done by Martins et al. (2022b). This process begins by constructing a histogram of the previous attention across intervals. Specifically, the signal is divided into $D$ linearly spaced bins, $\{d_1, \ldots, d_D\}$. The probability assigned to each bin, $p(d_j)$ for $j \in [D]$, is computed by integrating Eq. 10 over each bin interval using the trapezoidal rule:

$$p(d_j) \propto \sum_{h=1}^{H} \sum_{i=1}^{R} \int_{d_j} p_i^h(t). \qquad (15)$$

Finally, $T$ locations are sampled based on the resulting distribution, followed by the LTM update of §3.2. In our model, this sampling process is analogous to the phenomenon of non-local discontiguous "replay" in the brain whereby past experiences are reactivated for the purposes of consolidation (Carr et al., 2011; McNamee, 2024).

### 3.4. Model Architecture

We define the output context for the cross-attention layers of the video Q-former as a weighted sum of two components: the "local" vanilla video Q-former output context $Z_{\text{STM}} \in \mathbb{R}^{R \times d}$, which corresponds to the already present attention over the current chunk, and the "global" LTM context $Z_{\text{LTM}}$, which takes into account information from previous chunks as described in §3.2. The overall context is computed as:

$$\boldsymbol{Z} = \alpha \boldsymbol{Z}_{\text{STM}} + (1 - \alpha) \boldsymbol{Z}_{\text{LTM}}, \qquad (16)$$

where $\alpha$ is a weighting factor that balances the contribution of short-term and long-term memories. The video Q-formers considered in this work were trained with the $R$ tokens (or a subset) concatenated with the prompt tokens. In our approach, where the video context is divided into frame chunks processed by the long-term mechanism, it is necessary to aggregate information from all the $C$ chunks, in $\mathbb{R}^{C \times R}$, into a fixed set of $R$ tokens. To achieve this, we run the modified Video-LLaMA (Zhang et al., 2023b) and VideoChat2 (Li et al., 2024) models for each chunk and compute a running average of the embeddings as we process each chunk. The current video token embedding $\boldsymbol{E}_c \in \mathbb{R}^{R \times d}$, defined as $\boldsymbol{E}_c = \boldsymbol{W}_{\text{proj}} \boldsymbol{Z}_c$, is updated incrementally, enabling the model to handle arbitrarily long contexts without storing all embeddings of chunks in memory. The updated embedding is calculated as:

$$\bar{\boldsymbol{E}}_c = \frac{C-1}{C} \bar{\boldsymbol{E}}_{c-1} + \frac{1}{C} \boldsymbol{E}_c, \qquad (17)$$

Upon reaching the final chunk, the video token embeddings $\boldsymbol{E}$ are fed to the LLM that generates an answer. The full architecture diagram is shown in Fig. 1.

## 4. Experiments

In this section, we evaluate our proposed method on video question answering tasks, including multiple choice question answering (§4.2) and open-ended generation (§4.3).

### 4.1. Implementation Details

The Video-LLaMA (Zhang et al., 2023b) architecture that we adapt here with our LTM module was initially designed for short videos and employs a dual Q-Former architecture (Li et al., 2023a)—one for spatial and another for temporal feature extraction. We use the Video-LLaMA2-7B finetuned model,[3] leveraging EVA-CLIP's ViT-G/14 (Fang et al., 2022) as visual encoder and Vicuna 7B (Chiang et al., 2023) as our LLM.

We also adapt VideoChat2 (Li et al., 2024), a stronger short-video model equipped with a single video Q-Former and trained on extended instruction data. We use UMT-L (Li et al., 2023c), which captures both spatial and temporal dependencies but requires more memory. Thus, we use chunks with fewer frames. Finally, we follow Jiang et al. (2023) and use Mistral-7B (Jiang et al., 2023). Additional implementation details can be found in App. A.

In all our experiments, we approximate the integrals of Eqs. 10 and 11 using the trapezoidal rule with 1000 sampling points. For ∞-VIDEO with Video-LLaMA, we use 8 chunks of 256 frames with 1024 basis functions, except in the case of NeXT-QA (Xiao et al., 2021), where the total number of available frames is reduced. For ∞-VIDEO with VideoChat2, we use 8 chunks of 16 frames with $N = 256$ basis functions. We experiment with several values of $\alpha$ in Eq. 16. Full hyperparameters can be seen in App. A.2.

### 4.2. Multiple-Choice Question Answering

#### 4.2.1. COMPARISON WITH OTHER TRAINING-FREE METHODS

We consider systems of three kinds: (1) training-free approaches leveraging a GPT-4 backbone (Zhang et al., 2023a; Wang et al., 2024a;c); (2) models most similar to ours, which share the same underlying architecture, such as Video-LLaMA (Zhang et al., 2023b) and its training-free variants like MovieChat (Song et al., 2023) and MovieChat+ (Song et al., 2024); and (3) VideoChat2 (Li et al., 2024). For (2) and (3), we test ∞-VIDEO variants without LTM (corresponding to $\alpha = 1.0$) and those using uniform sampling and sticky memories with $\alpha = 0.9$.

**NeXT-QA.** We evaluate our models in NeXT-QA (Xiao et al., 2021), a dataset with questions and 5 multiple choice options about **short** videos with average duration of 44 seconds. As ∞-Video LLaMA can process a higher number of frames, we anticipate that using all the video information may lead to improvement over sub-sampling. For ∞-Video

---

[3] https://huggingface.co/DAMO-NLP-SG/Video-LLaMA-2-7B-Finetuned

*Table 1.* **Evaluation on Multiple Choice Datasets**. Evaluation accuracies on NeXT-QA (NeXT) (Xiao et al., 2021) and Egoschema subset (Ego) (Mangalam et al., 2023). * denotes results obtained by running the models on both datasets. † indicates models run on Egoschema but not on NeXT-QA. ◇ highlights models trained on NExT-QA. The remaining results are from Song et al. (2024) or Wang et al. (2024c). We bold the best-performing models for each category.

| Method | LLM | #Frames | NeXT | Ego |
|---|---|---|---|---|
| *Based on Proprietary LLMs* | | | | |
| LLovi (Zhang et al., 2023a) | GPT-4 | - | 67.7 | 61.2 |
| VideoAgent (Wang et al., 2024a) | GPT-4 | - | 71.3 | 60.2 |
| VideoTree (Wang et al., 2024c) | GPT-4 | - | **75.6** | **66.2** |
| *Video LLaMA-Based Models* | | | | |
| Video LLaMA* (Zhang et al., 2023b) | Vicuna-7B | 32 | 30.7 | 20.2 |
| MovieChat† (Song et al., 2023) | Vicuna-7B | 2048 | 34.4 | 41.6 |
| MovieChat+† (Song et al., 2024) | Vicuna-7B | 2048 | 35.2 | 37.4 |
| ∞-Video LLaMA (No LTM)* | Vicuna-7B | all/2048 | 37.6 | 40.8 |
| ∞-Video LLaMA (Uniform)* | Vicuna-7B | all/2048 | 37.5 | 42.6 |
| ∞-Video LLaMA (Sticky)* | Vicuna-7B | all/2048 | **41.1** | **46.8** |
| *VideoChat2-Based Models* | | | | |
| VideoChat2*◇ (Li et al., 2024) | Mistral-7B | 16 | **78.7** | 64.2 |
| ∞-VideoChat2 (No LTM)*◇ | Mistral-7B | 128 | 78.1 | 64.6 |
| ∞-VideoChat2 (Uniform)*◇ | Mistral-7B | 128 | 78.1 | 64.4 |
| ∞-VideoChat2 (Sticky)*◇ | Mistral-7B | 128 | 78.1 | **64.8** |

*Table 2.* **VideoMME results.** Baseline results are taken from (Fu et al., 2024). We bold the best performing models.

| Method | LLM | #Frames | Medium | Long | Avg |
|---|---|---|---|---|---|
| ST-LLM | Vicuna-7B | 64 | 36.8 | 31.1 | 37.9 |
| Video-LLaVA | Vicuna-7B | 8 | 38.0 | 36.2 | 39.9 |
| ShareGPT4Video 8B | - | 16 | 36.3 | 35.0 | 39.9 |
| Chat-UniVi-v1.5 | Vicuna-7B | 64 | **40.3** | 35.8 | 40.6 |
| Qwen-VL-Chat | Qwen-7B | 4 | 38.7 | 37.8 | 41.1 |
| VideoChat2 | Mistral-7B | 32 | 37.9 | 38.0 | 42.1 |
| ∞-VideoChat2 (no LTM) | Mistral-7B | 128 | 39.6 | 38.8 | 42.3 |
| ∞-VideoChat2 (uniform) | Mistral-7B | 128 | 40.0 | 38.8 | 42.4 |
| ∞-VideoChat2 (sticky) | Mistral-7B | 128 | 40.2 | **38.9** | **42.4** |

VideoChat2, though the improvements are less pronounced. We attribute this to the fact that VideoChat2 is inherently a stronger model, having been trained on a larger and more recent datasets, offering less room for improvement compared to the Video-LLaMA ∞-variants. However, gains are observed for both uniform sampling and sticky memories, with the latter showing superior performance. Moreover, despite having significantly fewer parameters, ∞-VideoChat2 demonstrates competitive results against proprietary LLMs based on ChatGPT-4.

### 4.2.2. EVALUATION ON VERY LONG VIDEOS

To emphasize the effectiveness of our approach on extended video content, we also present results on Video-MME (Fu et al., 2024), which features a diverse collection of lengthy videos, ranging up to 1 hour in duration. We compare our method against baseline models of similar size such as ST-LLM (Liu et al., 2024b), Video-LLaVA (Liu et al., 2023), ShareGPT4Video 8B (Chen et al., 2024a), Chat-UniVi-v1.5 (Jin et al., 2023) and Qwen-VL-Chat (Bai et al., 2023) as well as our ∞-VIDEO variants with $\alpha = 1$ and the base architecture VideoChat2 (Li et al., 2024).

In Tab. 2, we show the results for VideoMME. Although Video-LLaMA performs well on earlier datasets, its description-focused training dataset makes the performance as good as random guessing. Including it in the evaluation would detract from more relevant models. However, for the VideoChat2 category, sticky memories outperform others, followed by uniform sampling.

### 4.3. Long-Term Open-Ended Question Answering

We now investigate the performance of our models on open-ended question answering using the MovieChat-1K dataset (Song et al., 2023), a benchmark comprising long videos with an average duration of around 8 minutes. We compare our models with other baselines based on Vicuna-7B (Chiang et al., 2023), including VideoChat (Li et al., 2023b), Video-ChatGPT (Maaz et al., 2024), MovieChat (Song et al., 2023; 2024), MovieChat+ (Song et al., 2024). Furthermore,

LLaMA, we use all available frames in the videos, which we then split into chunks of up to 256 frames. In contrast, baseline models like Video-LLaMA (Zhang et al., 2023b) are limited to processing only 32 frames, while VideoChat2 achieves optimal performance for 16 frames, as shown in Li et al. (2024).

As shown in Tab. 1, ∞-Video LLaMA with sticky memories outperforms other approaches. This includes MovieChat+, which employs a heuristic to merge adjacent frames, fed to the video's Q-former cross-attention with question knowledge. In contrast, our model encounters the question only within the LLM prompt. Surprisingly, our method using uniform sampling of the continuous signal performs slightly worse than ∞-Video LLaMA without the LTM ($\alpha = 1$). For the VideoChat2 variants, we observe no significant performance increase. We attribute this to the in-domain nature of the evaluation, where the original model is already highly optimized for the given tasks.

**Egoschema.** We evaluate the training-free question-answering performance of our models on EgoSchema (Mangalam et al., 2023), a medium-length benchmark for ego-centric planning designed to test long-context video understanding with 3-minute videos.

Tab. 1 presents the results of our methods compared to strong parameter-free baselines. ∞-Video LLaMA, equipped with the LTM module ($\alpha = 0.9$) and using sticky memories, significantly outperforms both the uniform LTM variant, also for $\alpha = 0.9$ and the model without LTM ($\alpha = 1$), achieving a notable accuracy improvement of +6 points over the latter. A similar trend is observed with ∞-

*Table 3.* **MovieChat Results. Score** measures the overall answer score, **CI** stands for correctness of information, **DO** stands for detail orientation, and **CU** stands for contextual understanding. We bold the best results and underline the best within a category. We omit the temporal and consistency metrics due to the absence of the subset specific to these metrics. Except for Moviechat+, results were taken from (Song et al., 2023).

| Method | LLM | Number of Frames | Accuracy | Score | CI | DO | CU |
|---|---|---|---|---|---|---|---|
| Video Chat (Li et al., 2023b) | Vicuna-7B | 32 | 61.0 | 3.34 | 3.26 | 3.20 | 3.38 |
| Video-ChatGPT (Maaz et al., 2024) | Vicuna-7B | 100 | 44.2 | 2.71 | 2.48 | 2.78 | 3.03 |
| *Video LLaMA-Based Models* | | | | | | | |
| Video LLaMA (Zhang et al., 2023b) | Vicuna-7B | 32 | 51.4 | 3.10 | 3.30 | 2.53 | 3.28 |
| MovieChat (Song et al., 2023) | Vicuna-7B | 2048 | 67.8 | 3.81 | 3.32 | 3.28 | 3.44 |
| MovieChat+ (Song et al., 2024) | Vicuna-7B | 2048 | 66.4 | 3.67 | 3.70 | 3.30 | 3.62 |
| ∞-Video LLaMA (no LTM) | Vicuna-7B | 2048 | 68.0 | 3.76 | 3.72 | 3.33 | 3.71 |
| ∞-Video LLaMA (uniform) | Vicuna-7B | 2048 | 66.5 | 3.69 | 3.60 | 3.31 | 3.58 |
| ∞-Video LLaMA (sticky) | Vicuna-7B | 2048 | **72.2** | **3.88** | **3.89** | 3.47 | 3.79 |
| ∞-Video LLaMA (no STM uniform) | Vicuna-7B | 2048 | 62.4 | 3.75 | 3.36 | 3.38 | 3.52 |
| ∞-Video LLaMA (no STM sticky) | Vicuna-7B | 2048 | 59.2 | 3.68 | 3.30 | 3.30 | 3.44 |
| *VideoChat2-Based Models* | | | | | | | |
| VideoChat2 | Mistral-7B | 16 | 62.2 | 3.72 | 3.46 | 3.60 | 3.69 |
| ∞-VideoChat2 (no LTM) | Mistral-7B | 128 | 63.9 | 3.74 | 3.54 | 3.60 | 3.73 |
| ∞-VideoChat2 (uniform) | Mistral-7B | 128 | 64.1 | 3.73 | 3.54 | 3.60 | 3.75 |
| ∞-VideoChat2 (sticky) | Mistral-7B | 128 | 63.9 | 3.74 | 3.55 | 3.63 | 3.74 |
| ∞-VideoChat2 (no STM uniform) | Mistral-7B | 128 | 65.7 | 3.78 | 3.65 | 3.60 | 3.84 |
| ∞-VideoChat2 (no STM sticky) | Mistral-7B | 128 | 66.5 | 3.85 | 3.71 | **3.68** | **3.96** |

we evaluate our ∞-VIDEO variants with $\alpha = 0.9$ for both uniform sampling and sticky memories, as well as with $\alpha = 0$ (i.e., without STM). Following the standard evaluation method for open-ended questions, we prompt GPT-3.5 (OpenAI et al., 2024) for a yes/no answer prediction, a confidence score (0 to 5), and other qualitative metrics. The prompts are shown in App. A.3.

As shown in Tab. 3, our ∞-VIDEO LLaMA with sticky memories outperforms all models in its category, as well as VideoChat and Video-ChatGPT, across all metrics. It also surpasses MovieChat, which is designed for training-free long-context video understanding. In contrast, ∞-Video LLaMA with uniform sampling performs worse than both sticky memories and the model without LTM. Additionally, using only the LTM does not improve performance, highlighting that a weighted combination of STM and LTM yields the best results for the Video-LLaMA category. The same does not apply to the VideoChat2 category, where replacing the STM with the LTM yields top results. For $\alpha$ values other than 0, the performance of ∞-VIDEO with VideoChat2 remains nearly unchanged. Surprisingly, VideoChat2 underperforms compared to the Video LLaMA category despite being trained on a larger dataset. We hypothesize this is due to differences in the training datasets: Video LLaMA was trained on video descriptions with multiple sentences, encouraging more context, which increases the probability of correct predictions, while VideoChat2 was fine-tuned on concise datasets, favouring brief, open-ended predictions. Ablation studies on ∞-VIDEO with Video LLaMA are presented in App. B.1.

### 4.4. Qualitative Analysis

In Fig. 3, we show the continuous attention density map over the LTM in the final layer of the video Q-Former for the last chunk of ∞-VIDEO LLaMA. The example uses the *Interstellar* trailer, divided into 8 chunks of 256 frames each, with $\tau = 0.75$, $N = 1024$, and $\alpha = 0.9$. The bottom heatmap reveals that the continuous attention favours frames after $t = \tau$, where the sticky LTM exhibits peaks before this point. In contrast, the uniform LTM shows vanishing density as $t$ decreases, likely due to context contraction across chunks, which might explain the superior results for sticky memories.

In Fig. 4, we present the attention density as a function of the number of frames, with $\alpha = 0.9$, $N = 256$, and $\tau = 0.5$, for 3 chunks of 256 frames each, spanning 3 contraction steps. We also display 6 representative frames from high-density regions by identifying the top 26 frames within each interval and selecting the 6 non-redundant frames. The selected frames appear to align with visually striking or narratively significant scenes within the trailer, which suggests that ∞-VIDEO effectively might capture key moments in the video and discard irrelevant parts. For example, in the final interval, the region with the lowest attention density corresponds to the credits section of the video. We show a similar figure but for the uniform sampling in App. B.2.

## 5. Related Work

There are several recent advances in long-context video understanding (Li et al., 2023d; Liu et al., 2024a; Balazevic et al., 2024; Wang et al., 2024b; Shu et al., 2024; Ye

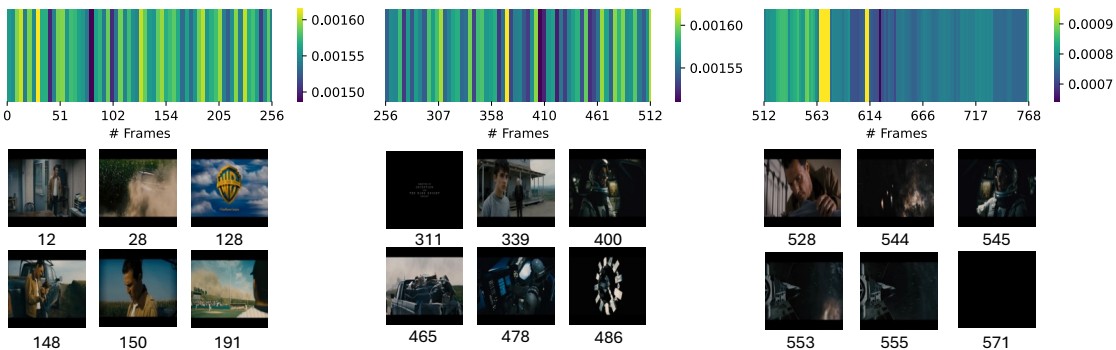

*Figure 3.* **(Top)** LTM attention density on the $[0, \tau]$ interval for the *Interstellar* trailer, using sticky memories in the final chunk of the ∞-Video LLaMA video Q-former's last layer. **(Bottom)** The same attention density map, extended over the full $t$ interval.

*Figure 4.* Highest continuous attention density frames selected using sticky memories in the *Interstellar* trailer for ∞-Video LLaMA across 3 chunks. **(Left)** Interval: $[0, \tau^2]$. **(Middle)** Interval: $(\tau^2, \tau]$. **(Right)** Interval: $(\tau, 1]$.

et al., 2024; Chen et al., 2024b), but few adapting transformers to leverage temporal information in a training-free setting. Closest to our work is Song et al. (2023; 2024), which extends Video LLaMA with memory consolidation by using a heuristic to merge similar frames, which alters frame embedding-level information. In contrast, our method retains embedding integrity, equipping the video Q-former's cross-attention with an LTM to efficiently process an arbitrary number of frames in one single pass over the video.

Moreover, works such as Zhang et al. (2024), Shu et al. (2024), and Chen et al. (2024b) address the challenge of long video contexts. However, these approaches involve fine-tuning or training models from scratch, which can be computationally expensive and time-intensive. Our approach, in contrast, enables the seamless adaptation of short video models to arbitrary long contexts without the need for sparse subsampling or discarding important information.

Our approach builds on continuous attention mechanisms, originally proposed by Martins et al. (2020), and later applied to image, speech, and natural language processing tasks (Farinhas et al., 2021; Martins et al., 2022a;b). We extend these ideas to video data by replacing the continuous softmax with the Gibbs PDF, which better replicates the discrete softmax used in the video Q-Former attention.

## 6. Conclusions

We introduced a lightweight extension to short-video vision LLMs, enabling arbitrary-long video understanding by augmenting the video Q-former's cross-attention mechanism with a long-term memory module that consolidates global information dynamically. Our approach adapts continuous attention to perform visual memory consolidation, allocating higher granularity to the most relevant frames. This ensures an efficient and focused representation of critical moments while maintaining scalability. Additionally, our method enables the sequential processing of videos with a single pass. Despite being training-free, our approach also paves the way for scalable long-context video understanding with transformers as spatio-temporal feature extractors.

Our work takes inspiration from cognitive and mechanistic theories of memory (re)consolidation in brains (Hardt et al., 2010; Preston & Eichenbaum, 2013; Ma et al., 2014), and a deeper integration with such theories may be pursued. For example, memory reactivation or "replay" is thought to be a key component of systems consolidation in "offline" states such as sleep. Our model could be extended to incorporate such with further training and schema-driven fine-tuning for the purposes of continual learning (Cai et al., 2024). Furthermore, as a neural architecture, our model goes beyond current brain models of episodic memory processing, which focus on discrete low-dimensional sequences of static images and relatively simple functionalities such as mem-

ory recall and event segmentation (Franklin et al., 2020; Chandra et al., 2025). Given our model's integration of rich and streaming visual input with flexible and sophisticated querying via text prompting, interpretability analyses of our architecture may provide insights regarding how episodic memory may be interrogated for complex inferences in the human brain (Tulving, 2002; Radvansky & Zacks, 2014).

## Impact Statement

We discuss the broader implications of our work, including ethical considerations and potential societal consequences. Our framework extends the capabilities of existing short-context multimodal language models, enabling them to process unbounded video contexts without requiring retraining. This is particularly relevant given concerns about the energy consumption of training large models (Strubell et al., 2019). However, we must also acknowledge the societal risks associated with video models, especially their potential use in privacy-violating surveillance. As our approach enables scaling to longer videos, there is a concern that it could be applied in undesirable domains. While current state-of-the-art models are often trained on datasets with documented biases, we intentionally focus on applications using standard benchmark applications, aiming to distance ourselves from harmful uses.

## Acknowledgments

We would like to thank Marcos Treviso, Giuseppe Attanasio, Sweta Agrawal, Chryssa Zerva and the SARDINE lab team for helpful discussions. This work was supported by EU's Horizon Europe Research and Innovation Actions (UTTER, contract 101070631), by the project DECOLLAGE (ERC-2022-CoG 101088763), by the Portuguese Recovery and Resilience Plan through project C645008882-00000055 (Center for Responsible AI), and by FCT/MECI through national funds and when applicable co-funded EU funds under UID/50008: Instituto de Telecomunicações.

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

## A. Implementation Details and Hyperparameters

### A.1. Additional Implementation Details

**Video LLaMA-Based Models.** As discussed in §3, Video-LLaMA (Zhang et al., 2023b) serves as one of the adapted models in this work. It employs a cascade of two Q-formers—one for spatial feature extraction and the other for temporal feature extraction—with the latter enhanced by our LTM integration. The video Q-former and projection layer parameters are consistent with the Video-LLaMA-2-7B-Finetuned model (Zhang et al., 2023b), which was fine-tuned using instruction-tuning data from MiniGPT-4 (Zhu et al., 2023), LLaVA (Liu et al., 2023), and VideoChat (Li et al., 2023b). For visual feature extraction, we utilize the ViT-G/14 encoder from EVA-CLIP (Fang et al., 2022), while spatial dependency features rely on the Q-former from BLIP-2 (Li et al., 2023a). This lightweight ViT facilitates efficient processing of a large number of frames per chunk. The LLM used in this model is Vicuna-7B (Chiang et al., 2023).

Atreja et al. (2024) and our empirical validation have shown limitations in respecting the format of the multiple-choice answer asked by the prompt, influencing the accuracy of this method in multiple-choice datasets. To address this, and for these datasets we follow the approach of (Song et al., 2023; 2024) and provide our modified model, ∞-Video LLaMA, exclusively with the questions in the prompt. Using LangChain (hwchase17, 2023), we calculate the similarity between ∞-Video LLaMA's open-ended responses and the given options, selecting the option that best aligns with the expected answer.

**VideoChat2-Based Models.** Building on Video-LLaMA, we also evaluated our methods using VideoChat2 (Li et al., 2024), a more advanced short-video model. Like Video-LLaMA, it incorporates a video Q-former, but it benefits from additional instruction-tuning data (Li et al., 2024). For visual encoding, we use UMT-L (Li et al., 2023c), which captures both spatial and temporal features specifically designed for video data. This model's higher memory requirements necessitated the use of smaller frame chunks in our experiments. For the LLM component, we used the stage-3 Mistral-7B version of VideoChat2, [4] which demonstrated the best performance in its original paper.

### A.2. Hyperparameters

We report in Tab. 4 the hyperparameters used in our experiments. For ∞-Video LLaMA and for NeXT-QA we use all the frames availables with variable number of chunks of 256 frames.

*Table 4.* Hyperparameters used in our ∞-variants for the different datasets.

| Parameter | NeXT-QA | | Egoschema | | VideoMME | MovieChat | |
| --- | --- | --- | --- | --- | --- | --- | --- |
| | ∞-Video LLaMA | ∞-VideoChat2 | ∞-Video LLaMA | ∞-VideoChat2 | ∞-VideoChat2 | ∞-Video LLaMA | ∞-VideoChat2 |
| # chunks | - | 8 | 8 | 8 | 8 | 8 | 8 |
| # frames | all | 16 | 256 | 16 | 16 | 256 | 256 |
| $N$ | 256 | 256 | 1024 | 256 | 256 | 1024 | 256 |
| $\tau$ | 0.75 | 0.75 | 0.75 | 0.75 | 0.5 | 0.75 | 0.75 |

### A.3. Evaluation

We further show in List. 1, 2, 3, 4 the prompts used for evaluation on open-ended question answering tasks for the Moviechat dataset.

*Listing 1.* ChatGPT-3.5 prompt for the overall accuracy and score metric.

```
"role": "system",
"content":
    "You are an intelligent chatbot designed for evaluating the correctness of generative outputs for question-answer pairs. "
    "Your task is to compare the predicted answer with the correct answer and determine if they match meaningfully. Here's how you can
        accomplish the task:"
    "------"
    "##INSTRUCTIONS: "
    "- Focus on the meaningful match between the predicted answer and the correct answer.\n"
    "- Consider synonyms or paraphrases as valid matches.\n"
    "- Evaluate the correctness of the prediction compared to the answer."
},
{
```

---

[4] https://huggingface.co/OpenGVLab/VideoChat2_stage3_Mistral_7B

```
"role": "user",
"content":
    "Please evaluate the following video-based question-answer pair:\n\n"
    f"Question: {question}\n"
    f"Correct Answer: {answer}\n"
    f"Predicted Answer: {pred}\n\n"
    "Provide your evaluation only as a yes/no and score where the score is an integer value between 0 and 5, with 5 indicating the
        highest meaningful match. "
    "Please generate the response in the form of a Python dictionary string with keys 'pred' and 'score', where value of 'pred' is  a
        string of 'yes' or 'no' and value of 'score' is in INTEGER, not STRING."
    "DO NOT PROVIDE ANY OTHER OUTPUT TEXT OR EXPLANATION. Only provide the Python dictionary string. "
    "For example, your response should look like this: {'pred': 'yes', 'score': 4.8}."
}
```

*Listing 2.* ChatGPT-3.5 prompt for the contextual understanding (CI) metric.

```
"role": "system",
"content":
    "You are an intelligent chatbot designed for evaluating the factual accuracy of generative outputs for video-based question-answer
        pairs. "
    "Your task is to compare the predicted answer with the correct answer and determine if they are factually consistent. Here's how
        you can accomplish the task:"
    "------"
    "##INSTRUCTIONS: "
    "- Focus on the factual consistency between the predicted answer and the correct answer. The predicted answer should not contain
        any misinterpretations or misinformation.\n"
    "- The predicted answer must be factually accurate and align with the video content.\n"
    "- Consider synonyms or paraphrases as valid matches.\n"
    "- Evaluate the factual accuracy of the prediction compared to the answer."
},
{
"role": "user",
"content":
    "Please evaluate the following video-based question-answer pair:\n\n"
    f"Question: {question}\n"
    f"Correct Answer: {answer}\n"
    f"Predicted Answer: {pred}\n\n"
    "Provide your evaluation only as a factual accuracy score where the factual accuracy score is an integer value between 0 and 5,
        with 5 indicating the highest level of factual consistency. "
    "Please generate the response in the form of a Python dictionary string with keys 'score', where its value is the factual accuracy
        score in INTEGER, not STRING."
    "DO NOT PROVIDE ANY OTHER OUTPUT TEXT OR EXPLANATION. Only provide the Python dictionary string. "
    "For example, your response should look like this: {''score': 4.8}."
```

*Listing 3.* ChatGPT-3.5 prompt for the detailed orientation (DO) metric.

```
"role": "system",
"content":
    "You are an intelligent chatbot designed for evaluating the detail orientation of generative outputs for video-based question-
        answer pairs. "
    "Your task is to compare the predicted answer with the correct answer and determine its level of detail, considering both
        completeness and specificity. Here's how you can accomplish the task:"
    "------"
    "##INSTRUCTIONS: "
    "- Check if the predicted answer covers all major points from the video. The response should not leave out any key aspects.\n"
    "- Evaluate whether the predicted answer includes specific details rather than just generic points. It should provide comprehensive
        information that is tied to specific elements of the video.\n"
    "- Consider synonyms or paraphrases as valid matches.\n"
    "- Provide a single evaluation score that reflects the level of detail orientation of the prediction, considering both completeness
        and specificity."
},
{
"role": "user",
"content":
    "Please evaluate the following video-based question-answer pair:\n\n"
    f"Question: {question}\n"
    f"Correct Answer: {answer}\n"
    f"Predicted Answer: {pred}\n\n"
    "Provide your evaluation only as a detail orientation score where the detail orientation score is an integer value between 0 and 5,
        with 5 indicating the highest level of detail orientation. "
    "Please generate the response in the form of a Python dictionary string with keys 'score', where its value is the detail
        orientation score in INTEGER, not STRING."
    "DO NOT PROVIDE ANY OTHER OUTPUT TEXT OR EXPLANATION. Only provide the Python dictionary string. "
    "For example, your response should look like this: {''score': 4.8}."
```

*Listing 4.* ChatGPT-3.5 prompt for the contextual understanding (CU) metric.

```
"role": "system",
"content":
    "You are an intelligent chatbot designed for evaluating the contextual understanding of generative outputs for video-based question
        -answer pairs. "
    "Your task is to compare the predicted answer with the correct answer and determine if the generated response aligns with the
```

```
            overall context of the video content. Here's how you can accomplish the task:"
    "------"
    "##INSTRUCTIONS: "
    "- Evaluate whether the predicted answer aligns with the overall context of the video content. It should not provide information
            that is out of context or misaligned.\n"
    "- The predicted answer must capture the main themes and sentiments of the video.\n"
    "- Consider synonyms or paraphrases as valid matches.\n"
    "- Provide your evaluation of the contextual understanding of the prediction compared to the answer."
},
{
"role": "user",
"content":
    "Please evaluate the following video-based question-answer pair:\n\n"
    f"Question: {question}\n"
    f"Correct Answer: {answer}\n"
    f"Predicted Answer: {pred}\n\n"
    "Provide your evaluation only as a contextual understanding score where the contextual understanding score is an integer value
            between 0 and 5, with 5 indicating the highest level of contextual understanding. "
    "Please generate the response in the form of a Python dictionary string with keys 'score', where its value is contextual
            understanding score in INTEGER, not STRING."
    "DO NOT PROVIDE ANY OTHER OUTPUT TEXT OR EXPLANATION. Only provide the Python dictionary string. "
    "For example, your response should look like this: {''score': 4.8}."
```

# B. Additional Experiments

## B.1. Ablation Studies on ∞-Video LLaMA

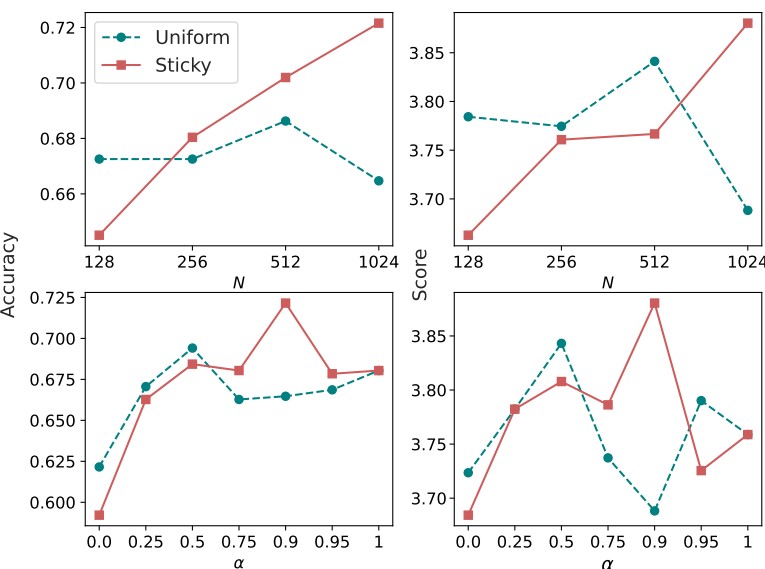

*Figure 5.* **Ablation studies on the MovieChat dataset:** Evaluation of accuracy and score metrics for various values of the number of basis functions $N$ and the contribution of long-term memory $\alpha$.

In this section, we conduct ablation studies on ∞-Video LLaMA on the MovieChat dataset. We ablate three hyperparameters: the percentage of the long-term memory used $\alpha$, the number of basis functions $N$ and the sampling method. We explore $\alpha \in \{0, 0.25, 0.5, 0.75, 0.95, 1\}$, covering the full spectrum from exclusively using the LTM ($\alpha = 0$) to exclusively using the STM ($\alpha = 1$). Additionally, we vary the number of basis functions with $N \in \{128, 256, 512, 1024\}$ to evaluate the impact of this parameter on performance and vary the sampling as either uniform or sticky.

Fig. 5 presents the accuracy and score provided by ChatGPT 3.5 as functions of the explored hyperparameters. The results reveal a general trend of increasing accuracy and score with $\alpha$, up to a certain point, after which a slight decline is observed as $\alpha$ approaches 1. This trend is somewhat explained by the inherent variability in ChatGPT's outputs. Additionally, both accuracy and score tend to be higher for sticky memories compared to uniform sampling from $\alpha = 0.75$ onward, whereas uniform sampling demonstrates superior performance for lower values of $\alpha$. Another observed trend is that as the number of basis functions increases, both metrics improve. However, for uniform sampling, performance slightly decreases beyond $N = 512$.

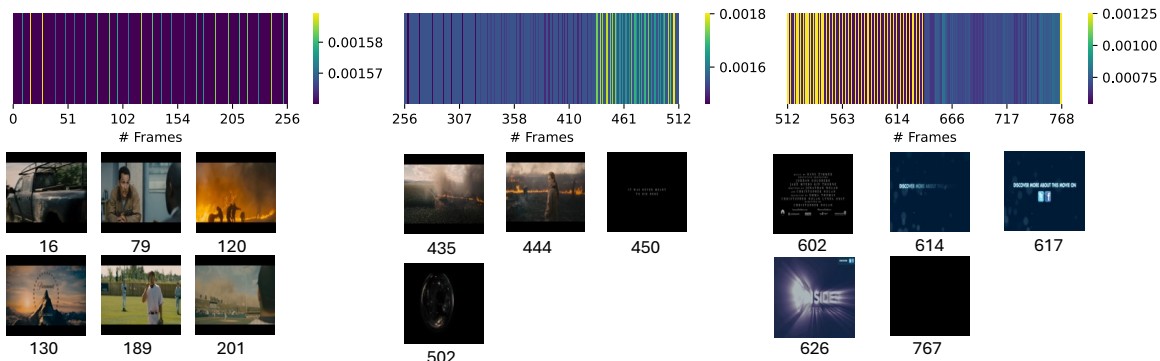

*Figure 6.* Highest continuous attention density frames selected using uniform memories in the *Interstellar* trailer for ∞-Video LLaMA across 3 chunks. **(Left)** Interval: $[0, \tau^2]$. **(Middle)** Interval: $(\tau^2, \tau]$. **(Right)** Interval: $(\tau, 1]$.

## B.2. Qualitative Analysis

In Fig. 6, we illustrate the attention density as a function of the number of frames for the uniform sampling LTM configuration, using $\alpha = 0.9$, $N = 256$, and $\tau = 0.5$. This analysis spans 3 chunks of 256 frames each, corresponding to 3 contraction steps. Additionally, we showcase representative frames from high-density regions by identifying the top 10 frames in each interval and selecting non-redundant examples.

The results reveal that, while the model effectively identifies key moments during the first two contraction steps, it disproportionately focuses on the credits scene in the final step. This behaviour contrasts with the results in Fig. 4, where such focus is avoided.

We leave in Fig. 7 additional examples of predictions of our modified ∞-Video LLaMA. We divide the video into 8 chunks of 256 frames with N=1024, $\tau = 0.75$ and $\alpha = 0.9$ both for uniform sampling and sticky memories.

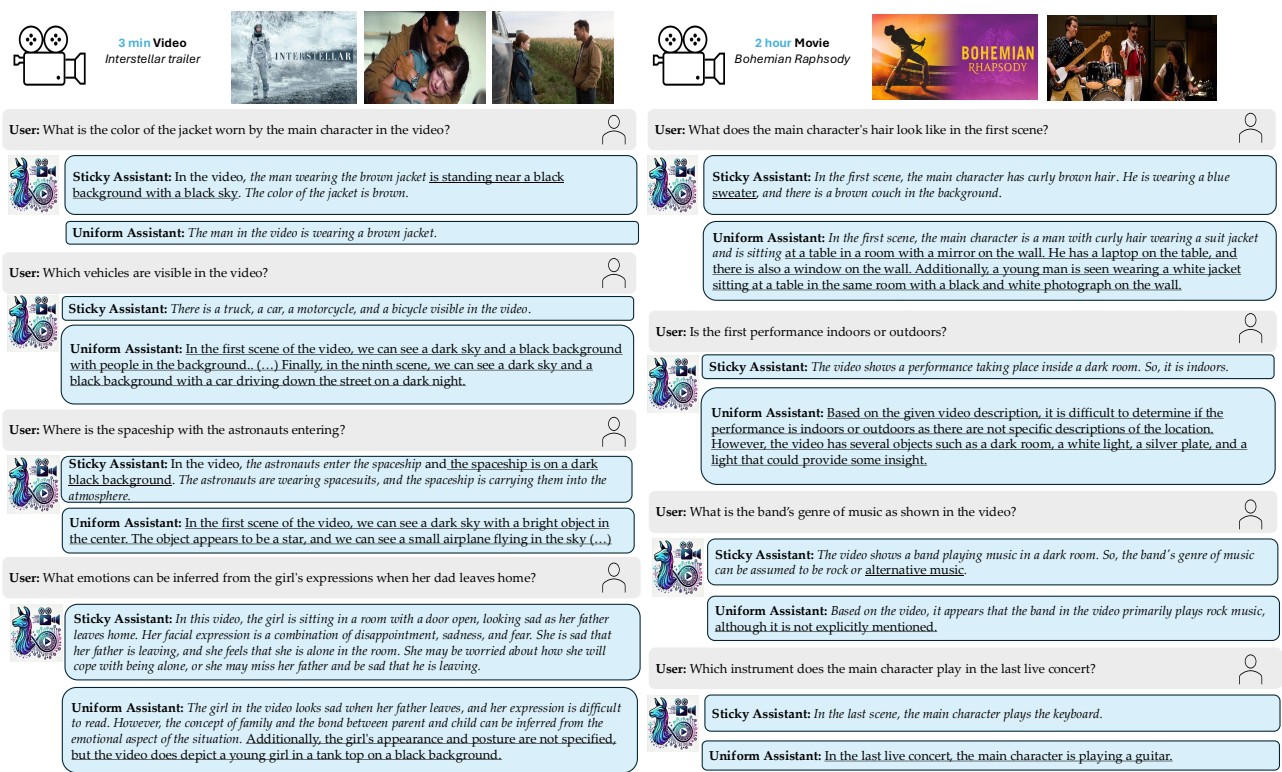

*Figure 7.* Examples of ∞-Video LLaMA answers with uniform sampling and sticky memories for short and ultra-long videos. Italicized corresponds to the correct answer while underlined corresponds to the wrong answer or hallucination.

