# OpenReview forum: "$\infty$-Video: A Training-Free Approach to Long Video Understanding via Continuous-Time Memory Consolidation"
_ICML.cc/2025/Conference — ICML 2025 poster_

### Official Review · Reviewer_j2dG · 2025-02-16

**Overall Recommendation:** 3

**Summary:**

This paper proposes a training-free approach for long-form video understanding tasks. The method introduces a memory bank that integrates modality projectors (Q-formers), which combine short-term and long-term memory for more efficient video processing. By leveraging this architecture, the framework can handle videos without requiring additional training. Experiments conducted on two different video benchmarks, Video-LLama and VideoChat2, show that the method provides clear improvements in long-form video understanding tasks, particularly in video question-answering.

**Claims And Evidence:**

Yes, the claims made in the submission are supported by clear and convincing evidence from the experimental results. The authors demonstrate that their approach, which integrates a memory bank and modality projectors, effectively improves the performance of video understanding models on standard benchmarks.

**Essential References Not Discussed:**

The paper ignored some recent state-of-the-art video-language models, such as Apollo[1], Video-LLaMA[2], and Llava-OneVision[3]. These models have made significant contributions to the field and could provide useful comparisons for the proposed method.

[1] Zohar, Orr, et al. "Apollo: An exploration of video understanding in large multimodal models." arXiv preprint arXiv:2412.10360 (2024).

[2] Zhang, Hang, Xin Li, and Lidong Bing. "Video-llama: An instruction-tuned audio-visual language model for video understanding." arXiv preprint arXiv:2306.02858 (2023).

[3] Li, Bo, et al. "Llava-onevision: Easy visual task transfer." arXiv preprint arXiv:2408.03326 (2024).

**Experimental Designs Or Analyses:**

The experimental settings are clear. The paper presents well-defined experiments with appropriate baselines and evaluation metrics. However, one concern is that the authors seem to use relatively old models as the base for their comparisons.

**Methods And Evaluation Criteria:**

Yes, the proposed training-free method has the impact on application side, addressing the challenge of long-form video understanding without requiring additional training.

**Other Comments Or Suggestions:**

NA

**Other Strengths And Weaknesses:**

Strengths:
1. The proposed method uses a memory bank to address long context has been proven both efficient and effective for long-form video understanding.
2. The paper extended the framework from MovieChat by introducing Continuous-Time Memory Consolidation, which is better than uniformed sampling. The continuous-time strategy helps capture more relevant video segments and improves overall efficiency.
3. The experiments are solid, utilizing multiple widely-used datasets, which makes the results more robust and generalizable to various tasks in video understanding.


Weaknesses: See Question Section.

**Questions For Authors:**

1. Is there any efficiency analysis provided in terms of computational resources, such as FLOPs or GPU memory cost for the proposed method? For example, a visualization like the figure 1 in MovieChat paper would be better.
2. In Table 3, why do video-llama-based models perform best in sticky settings while VideoChat2  are in no short-term memory sticky?
3. Is it possible to set alpha as a dynamic parameter instead of keeping it fixed at 0.9/1.0 statically? A dynamic alpha might allow more flexible adaptation based on varying video contexts.
4. The baseline models used in the paper (Video-LLama and VideoChat2) are quite old (from spring and summer 2023). Given the rapid advancements in video understanding, there are newer and more powerful models available. It would be better to consider including some of these models, such as Llava-OneVision[1], Qwen2-VL[2], or DeepSeek-VL[3], in future experiments?

[1] Li, Bo, et al. "Llava-onevision: Easy visual task transfer." arXiv preprint arXiv:2408.03326 (2024).

[2] Wang, Peng, et al. "Qwen2-vl: Enhancing vision-language model's perception of the world at any resolution." arXiv preprint arXiv:2409.12191 (2024).

[3] Lu, Haoyu, et al. "Deepseek-vl: towards real-world vision-language understanding." arXiv preprint arXiv:2403.05525 (2024).

**Relation To Broader Scientific Literature:**

The proposed method further extends the memory-based long-form video understanding framework by incorporating more dynamic resource allocation. By augmenting the memory model with a continuous-time mechanism, the approach provides a way to process long-form videos more effectively.

**Theoretical Claims:**

There is quite a little theoretical part in this paper. The authors primarily focus on the empirical performance of their method, with limited theoretical exploration of how the continuous-time memory mechanism interacts with the video understanding tasks.

---

> ### Author Rebuttal · Authors · 2025-03-31
>
> Thank you for your positive review and suggestions. We are happy that you found our method to be both efficient and effective for long-form video understanding, and our experiments solid. We address your concerns about our paper below.
>
> > “Is there any efficiency analysis provided in terms of computational resources, such as FLOPs or GPU memory cost for the proposed method? For example, a visualization like the figure 1 in MovieChat paper would be better.”
>
> We appreciate the suggestion and refer to our response to reviewer iEVz, where we present additional experiments that analyze the computational overhead. These experiments include the time consumption of the LTM module and the impact of sampling more frames. Our results demonstrate that our method sustains a constant memory footprint regardless of the number of frames, with only a slight increase in inference time. Additionally, we observe that increasing the number of basis functions leads to a much smaller rise in memory usage compared to the baseline's growth with additional frames. We will add these experiments to the appendix.
>
> > “In Table 3, why do video-llama-based models perform best in sticky settings while VideoChat2 are in no short-term memory sticky?”
>
> This is indeed intriguing and we asked ourselves the same question. We speculate that VideoChat2-based models tend to prioritize global features, whereas Video-LLaMA tends to focus on local features. Both benefit from sticky memories in open-ended generation tasks, as it helps maintain long-term coherence and consistency across outputs, regardless of whether the model emphasizes global or local context.
>
> > “Is it possible to set alpha as a dynamic parameter instead of keeping it fixed at 0.9/1.0 statically? A dynamic alpha might allow more flexible adaptation based on varying video contexts.”
>
> Thank you for the thoughtful suggestion. We agree that using a fixed $\alpha$ may not always be ideal, as video content with different temporal dynamics might benefit from varying weightings. While we chose a fixed $\alpha$ to simplify the current experiments and establish a baseline, we recognize the potential advantages of making $\alpha$ dynamic. Adapting $\alpha$ based on video characteristics (as mentioned by reviewer **aCqS**), such as shot transitions or object persistence, could indeed improve performance and generalization. We plan to explore this dynamic approach in future work, and we will mention it in the discussion section.
>
> > “The baseline models used in the paper (Video-LLama and VideoChat2) are quite old (from spring and summer 2023). Given the rapid advancements in video understanding, there are newer and more powerful models available. It would be better to consider including some of these models, such as Llava-OneVision[1], Qwen2-VL[2], or DeepSeek-VL[3], in future experiments?”
>
> While we recognize the rapid advancements in video understanding and the availability of more powerful models, our primary goal in this work was not to develop a SOTA model but to demonstrate the effectiveness of incorporating a biologically inspired LTM in a fair, training-free setting. The baseline models we used (Video-LLama and VideoChat2) were chosen for their relevance to our approach given that their Q-former based architecture enables a training-free approach, but we agree that including newer models like Llava-OneVision, Qwen2-VL, or DeepSeek-VL in future experiments (which would require additional training) may provide valuable comparisons. Our method is highly general and can be integrated into stronger VLMs moving forward. We will highlight this in the discussion section and suggest future work to explore the incorporation of our LTM approach into state-of-the-art models.

---

> > ### Comment · Reviewer_j2dG · 2025-04-02
> >
> > Thank you for the detailed rebuttal. While the response does clarify several technical points, I remain concerned about the limited empirical validation of the proposed training-free method on stronger or more recent base models. As the method claims to be training-free, it should be feasible to demonstrate broader applicability without significant computational overhead. The lack of such evidence weakens the generality and practical value of the approach. Therefore, I will maintain my original score.

---

> > > ### Author Response · Authors · 2025-04-05
> > >
> > > Thank for your response. We are happy that our answer clarified the technical points you were concerned about in your review. Regarding the empirical validation of our proposed training-free method: we understand you would like to see experiments on stronger or more recent base models beyond the two models we already experimented in: Video-LLaMA and VideoChat2. However, we maintain that doing so **while keeping the model training-free** is not straightforward. Our continuous attention framework requires spatio-temporal modules based on transformers to map the embeddings from the ViT to tokens understandable by LLMs, which models Video-LLaMA and VideoChat2 satisfy with the video Q-formers. Unfortunately this is not possible with other models such as the ones you mention in your review, Apollo, Llava-OneVision, Qwen2-VL, or DeepSeek-VL as these use projection or pooling layers to do this mapping . Integration of our continuous memories with those models would require expensive fine-tuning with substantial computational overhead, which goes beyond the scope of our paper and surpasses our limited computational budget. We believe the experiments with Video-LLaMA and VideoChat2 on 3 different benchmarks serve the main purpose of our paper, which is to show that our memory consolidation mechanism using continuous attention is a simple add-on component that improves model capabilities with minimal computational effort.

---

### Official Review · Reviewer_aCqS · 2025-03-11

**Overall Recommendation:** 3

**Summary:**

The paper presents a method for long video understanding through a continuous long-term memory (LTM) consolidation mechanism. In their approach, the authors propose a continuous-time attention mechanism that leverages the Gibbs density function to obtain a continuous-time query-key similarity function. Using this similarity function, they update the video Q-former's attention mechanism to make it utilize continuous-time LTM, with particular attention on the most relevant parts of input videos given the input prompts, for efficient understanding of long videos. The authors perform relevant experiments to evaluate their approach on multiple video understanding datasets and compare against baseline methods.

**Claims And Evidence:**

The central claim of chain-of-shot prompting improving video understanding performance is backed up by experimental results.

**Essential References Not Discussed:**

While not a domain expert, I did not find any major missing references in my search.

**Experimental Designs Or Analyses:**

1. As the authors mention, one key benefit of using basis functions over processing individual video frames is that fewer basis functions are needed to represent the information in the raw frames. This leads to more compressed representations (Lines 145-147), which is particularly useful for processing long videos. However, in their experiments, the authors use 256 frames in each chunk and 1024 basis functions (Line 238, col 2) for VideoLLaMA and 16-frame chunks and 256 basis functions for VideoChat2 (Line 241, col 2). How do these experimental designs reconcile with the assumption of needing fewer basis functions than video frames? Or are the authors implying that the number of basis functions is fewer than the *total* number of frames across all the chunks (e.g., 8 $\times$ 256 = 2048 > 1024)? But even with that assumption, for VideoChat2, we get 8 $\times$ 16 = 128 < 256. Could the authors please explain this discrepancy?

2. For the choice of the weighting factor $\alpha$ between short- and long-term memories (Eqn. 16), the authors report experiments on the MovieChat dataset and choose the best value of $\alpha$ from those experiments. However, it is unclear how generalizable that value is to other videos or datasets. Also, is it fair to assume that $\alpha$ should be constant for all kinds of videos, or is there scope to make $\alpha$ context- and category-aware, e.g., depending on how often shots are cut in the video, how often background scenes and foreground objects appear and reappear in the video, etc.?

3. Have the authors considered the effect of any type of noise when determining relevant locations for the sticky memory? For example, if the signal is slightly perturbed (additive noise), how would the probability function (Eqn. 15) be affected? Understanding the noise characteristics is not essential to determine the utility of the sticky memory procedure, and my rating would not depend on the authors' response to this question. However, the noise characteristics become relevant when considering that video signals can easily be corrupted by transmission, compression, or even adversarial noises.

**Methods And Evaluation Criteria:**

The proposed ideas of processing long-term memory in a continuous fashion to improve the attention mechanism of video Q-formers for long videos and sampling relevant parts of the memory at higher granularity to improve Q&A performance make sense as an approach for making long video understanding feasible and efficient.

**Other Comments Or Suggestions:**

In Eqn. 2, $Q \in \mathbb{R}^{R \times d}, K \in \mathbb{R}^{L \times d}, V \in \mathbb{R}^{L \times d} \Rightarrow QK^\top \in \mathbb{R}^{R \times L} \Rightarrow QK^{\top}V \in \mathbb{R}^{R \times d}$. This implies that $Z \in \mathbb{R}^{R \times d}$, but the authors have stated $Z \in \mathbb{R}^{L \times d}$ in both Lines 116 and 121. Is this a typo?

**Other Strengths And Weaknesses:**

N/A

**Questions For Authors:**

Please refer to the comments in previous sections.

**Relation To Broader Scientific Literature:**

With the development of language models for video understanding, the paper's contribution is relevant and timely, and it establishes new baselines for future video understanding models. It will likely interest the broader scientific communities working on language models, video understanding, and their intersections.

**Theoretical Claims:**

Not applicable - the paper presents experimental findings to justify the proposed approach.

---

> ### Author Rebuttal · Authors · 2025-03-31
>
> Thank you for your positive review and suggestions. We address your concerns about our paper below.
>
> > “As the authors mention, one key benefit of using basis functions over processing individual video frames is that fewer basis functions are needed to represent the information in the raw frames. This leads to more compressed representations (Lines 145-147), which is particularly useful for processing long videos. However, in their experiments, the authors use 256 frames in each chunk and 1024 basis functions (Line 238, col 2) for VideoLLaMA and 16-frame chunks and 256 basis functions for VideoChat2 (Line 241, col 2). How do these experimental designs reconcile with the assumption of needing fewer basis functions than video frames? Or are the authors implying that the number of basis functions is fewer than the total number of frames across all the chunks (e.g., 8x256 = 2048 > 1024)? But even with that assumption, for VideoChat2, we get 8x16 = 128 < 256. Could the authors please explain this discrepancy?”
>
> For Video LLaMA, where the number of total frames is large (2048), the basis functions were set to half this total number of frames, ensuring a compressed yet effective representation. In the case of VideoChat2, we used 256 basis functions with $L=128$ frames since the number of frames supported by VideoChat2 is much smaller. Increasing the number of basis functions is computationally lighter than increasing chunk size (as shown in the first two Tables in **iEVz** response) and it improves the multivariate step regression fit. Larger chunk sizes caused memory issues for VideoChat2, but increasing basis functions did not, allowing for better fitting. This strategy balances memory efficiency and model accuracy without overloading computational resources. This tradeoff will be further discussed in the appendix.
>
> > “For the choice of the weighting factor alpha between short- and long-term memories (Eqn. 16), the authors report experiments on the MovieChat dataset and choose the best value of α from those experiments. However, it is unclear how generalizable that value is to other videos or datasets. Also, is it fair to assume that α should be constant for all kinds of videos, or is there scope to make α context- and category-aware, e.g., depending on how often shots are cut in the video, how often background scenes and foreground objects appear and reappear in the video, etc.?”
>
> Thank you for the insightful suggestion. While we intentionally used a fixed $\alpha$ to establish a baseline and simplify experiments, we agree that an adaptive approach—adjusting $\alpha$ based on video characteristics like shot changes or object permanence—might be an interesting idea to improve generalization in future work. We will mention it in the discussion section.
>
> > “Have the authors considered the effect of any type of noise when determining relevant locations for the sticky memory? For example, if the signal is slightly perturbed (additive noise), how would the probability function (Eqn. 15) be affected? Understanding the noise characteristics is not essential to determine the utility of the sticky memory procedure, and my rating would not depend on the authors' response to this question. However, the noise characteristics become relevant when considering that video signals can easily be corrupted by transmission, compression, or even adversarial noises.”
>
> This is an interesting question. The impact will depend on the type of noise and on its smoothness in the time dimension. If the signal is perturbed with additive noise, the Gibbs transformation in Eq. 10 will lead to a modified density, which in turn will change the locations for the sticky memory as for Eq. 15. Since the Gibbs transformation is continuous and well-behaved, the effect in the memory representations should be smooth. However, to better understand the practical effect of noise in the memory a careful empirical analysis needs to be conducted.
>
> > “In Eqn. 2, $Q \in \mathbb{R}^{R \times d}$, $K \in \mathbb{R}^{L \times d}$, $V \in \mathbb{R}^{L \times d}$ (...) implies that $Z \in \mathbb{R}^{R \times d}$, but the authors have stated $Z \in \mathbb{R}^{L \times d}$ in both Lines 116 and 121. Is this a typo?”
>
> Yes, this is indeed a typographical error. We appreciate your careful review and will correct it accordingly. Thank you for bringing this to our attention.

---

### Official Review · Reviewer_SJwR · 2025-03-14

**Overall Recommendation:** 2

**Summary:**

This paper introduces a long-term memory (LTM) consolidation mechanism from $\infty$-Former, and a long-video LLM (Language Model) that requires no additional training based on existing short-video LLMs. Experimental results show that this approach significantly improves performance on long-video benchmarks. Furthermore, the use of Gibbs density in LTM for improved PDF sampling, the establishment of continuous-time memory and the use of stick memory sampling strategy are also introduced. The ablation study shows the effects of the proposed sticky memory and LTM. The contribution to understanding brain memory is also highlighted.

**Claims And Evidence:**

Strengths:

1. The introduction of LTM has effectively improved the performance of long video understanding, which has been strongly validated in the experiment section.
2. The claim of the generalizability from short video to long video understanding is well proved in methods and experiments.


Weaknesses:

1. The authors claim that the proposed PDF based on Gibbs density is much powerful than the Gaussian model in L96-99, which lacks neither theoretical proof nor experimental results.

**Essential References Not Discussed:**

Most related works are well cited.

**Experimental Designs Or Analyses:**

Strengths:

1. The ablation study on different setting is well conducted, proving the effects of each proposed module.
2. The visualization of the LTM attention density clearly shows the importance of sticky memory.



Weaknesses:

1. Though the method is training-free, there is still computational overhead should be considered. That includes the time consumption of the proposed LTM module, the extra time brought by sampling more frames from the video and encoding them with ViT encoder.
2. The experimental results shown in Tab. 1 and Tab. 2 demonstrate limited improvement for VideoChat2, and the authors attribute this to the optimization of the original model. Recent models have significantly outperformed VideoChat2 on various video understanding benchmarks, raising concerns about the applicability of the proposed method on stronger models.
3. The baseline models used for comparison are too outdated. Most of these models were produced in early 2024 or in 2023, and their performance is far inferior to current models.

**Methods And Evaluation Criteria:**

Strengths:

1. The idea of introducing the brain's working mechanism into video LLMs through structural  design rather than training another model is novel and reasonable.
2. The use of representing L signals through N basis functions significantly reduces context length, which contributes to longer video understanding.



Weaknesses:

1. The proposed method heavily relies on the video Q-Former. However, recent models seldom use Q-Former, as previous research [Tong'24] has demonstrated the flaws in Q-Former. Even the latest works of Video-LLaMA series [Zhang'25] and VideoChat series [Li'25] mentioned in the paper, though as the concurrent works, do not use Q-Former. Over-reliance on Q-Former may affect the method's generalizability and limit the possibilities of applying the method to stronger baselines.


\[Tong'24\]: Tong et. al, Cambrian-1: A Fully Open,Vision-Centric Exploration of Multimodal LLMs, NeurIPS 2024

\[Li'25\]: Li et. al, VideoChat-Flash: Hierarchical Compression for Long-Context Video Modeling

\[Zhang'25\]: Zhang et. al, VideoLLaMA 3: Frontier Multimodal Foundation Models for Image and Video Understanding

**Other Comments Or Suggestions:**

1. The last 'a' of Video-LLaMA in L288 should be capitalized.

2. The reference format should be unified and refined. Most names that should be capitalized are not capitalized properly. Some of the references are cited with url while others are not. Some arxiv papers are cited with arxiv IDs, some are cited with url and others are cited without additional information. Some conference names are annotated with abbreviations, while others are not.

**Other Strengths And Weaknesses:**

Strengths:

1. The creative combination and analysis of video understanding and brain memory may contribute to neuroscience



Weaknesses:

1. The limited performance of the model might be a severe bottleneck for real-world application of the method.

**Questions For Authors:**

1. Could the authors give proof of the advantage of using the complex Gibbs density for the PDF instead of the original Gaussian model?
2. Considering the shortcomings of Q-Former, could the authors talk about the generalizability to other models without video Q-Former?
3. To better prove the effectiveness of the proposed method, could the authors provide a detailed analysis of time consumption for inferring, including the time used for pre-calculating the ridge regression $F^T(F F^T+\lambda I)^{-1}$, the extra time brought by the use of basis functions $\psi$, the extra time of using LTM, and most importantly the extra time brought by sampling much more frames and encoding them with ViT encoder? It's recommended to compare the time consumption with LLM forward time.
4. Could the authors give more detailed analysis on the reason of limited improvement on the strong baseline VideoChat2?
5. Could the authors conduct the experiments based on stronger baselines to prove the generalizability of the proposed method and provide more comparisons to recent stronger models?
6. Could the author fix the typos and reference format issues mentioned above?

**Relation To Broader Scientific Literature:**

This paper utilizes the continuous attention [Martins'20] and $\infty$-former [Martins'22] inspired from cognitive and mechanistic theories of memory [Hardt'09, Weilbächer&Gluth'13, Ma'14] to enhance video models based on VideoChat2 [Li'23] and Video-LLaMA [Zhang'23]. The paper connects the field of neuroscience and video LLM and enables new understanding of neuroscience through the analysis on the mechanism of video LLM memory.

\[Martins'20\]: Martins et. al, Sparse Continuous Distributions and Fenchel-Young Losses, JMLR 2022

\[Martins'22\]: Martins et. al, $\infin$-former: Infinite Memory Transformer, ACL 2022

\[Hardt'09\]: Hardt et. al, A Bridge Over Troubled Water: Reconsolidation as a Link Between Cognitive and Neuroscientific Memory Research Traditions, Annual Review of Psychology

\[Weilbächer&Gluth'13\]: Weilbächer and Gluth, The Interplay of Hippocampus and Ventromedial Prefrontal Cortex in Memory-Based Decision Making, Current Biology

\[Ma'14\]: Ma et. al, Changing concepts of working memory, Nature Neuroscience

\[Li'23\]: Li et. al, MVBench: A Comprehensive Multi-modal Video Understanding Benchmark, CVPR 2024

\[Zhang'23\]: Zhang et. al, Video-LLaMA: An Instruction-tuned Audio-Visual Language Model for Video Understanding, EMNLP 2023 Demo

**Theoretical Claims:**

The theoretical introduction of continuous LTM is well presented.

---

> ### Author Rebuttal · Authors · 2025-03-31
>
> Thank you for your review and suggestions. We are happy that you found our idea novel, the generalizability from short video to long video understanding well proved in both methods and experiments, and the ablation studies well conducted. We understand your main concerns about our paper and address them below. We hope that our answers clarify and alleviate your concerns.
>
> > “The authors claim that the proposed PDF based on Gibbs density is much powerful than the Gaussian model in L96-99, which lacks neither theoretical proof nor experimental results.”
>
> We agree that our claim will benefit from concrete evidence. In fact, at an initial stage of our project, we experimented with the Gaussian model but we abandoned it since its unimodal nature limits its ability to capture complex distributions, leading to poor results. Our method extends Video Q-Former's cross-attention using the same learned projections trained with softmax, making Gibbs density a more natural fit. Nevertheless, we conducted additional experiments comparing both approaches, confirming that Gibbs improves performance. We also observed that most of the responses in open-ended MovieChat-1K degraded significantly under the Gaussian model. We will add this to the appendix.
>
> |Density|Method|Acc|Score|CI|DO|CU|
> |-|-|-|-|-|-|-|
> |Gibbs|Ours (no LTM)|68.0|3.76|3.72|3.33|3.71|
> |Gibbs|Ours (unif.)|66.5|3.69|3.60|3.31|3.58|
> |Gibbs|Ours (sticky)|**72.2**|**3.88**|**3.89**|**3.47**|**3.79**|
> |Gibbs|Ours (no STM unif.)|62.4|3.75|3.36|3.38|3.52|
> |Gibbs|Ours (no STM sticky)|59.2|3.68|3.30|3.30|3.44|
> |Gaussian|Ours (unif.)|46.3|2.98|3.60|2.31|3.16|
> |Gaussian|Ours (sticky)|38.4|2.76|3.51|2.11|3.12|
> |Gaussian |Ours (no STM unif.)|46.9|3.06|3.54|2.40|3.13|
> |Gaussian |Ours (no STM sticky)|38.6|2.77|3.43|2.15|3.13|
>
> > “Though the method is training-free, there is still computational overhead should be considered.”
>
> We refer to our response to reviewer iEVz, where we present additional experiments analyzing the computational overhead, including the time consumption of the LTM module, the impact of sampling more frames, and the encoding time with the ViT encoder. Our results demonstrate that our method sustains a constant memory footprint regardless of the number of frames, with only a slight increase in inference time. Additionally, we observe that increasing the number of basis functions leads to a much smaller rise in memory usage compared to the baseline's growth with additional frames. We hope that these experiments alleviate your concern.
>
> > “The proposed method heavily relies on the video Q-Former. However, recent models seldom use Q-Former (...)”; “The baseline models used for comparison are too outdated (...)”
> We chose Q-Former because it allows for seamless, training-free integration of our biologically inspired continuous attention component. This is possible because the video Q-Former is designed to process sequences of frame representations using a cross-attention module, which enables our method to augment it with a LTM based on continuous cross-attention. However, our LTM, if we want to go beyond training-free, is not inherently tied to Q-Former and could be used with other architectures. Our primary goal is to provide a proof of concept  incorporating a biologically inspired LTM in a fair, training-free setting. While SOTA models continue to evolve rapidly, our focus was on conducting an apples-to-apples comparison to validate the benefits of our approach. Our method is highly general and can be integrated into stronger VLMs in the future. Unlike our training-free approach, this would require additional learning. We will provide insights in the discussion section. Future work can then build on our findings by incorporating the LTM into the strongest VLMs available.
>
> > “Tab. 1 and Tab. 2 demonstrate limited improvement for VideoChat2 (...). Recent models have significantly outperformed VideoChat2 (...) give more detailed analysis on the reason of limited improvement (...)”
>
> This is an interesting question. The results in Tab. 1-2 are for multiple-choice datasets, where VideoChat2 models directly incorporate answer options into the prompt, making responses more constrained. In contrast, models like Video-LLaMA generate open-ended responses before selecting the closest option match with LangChain, as explained in the Appendix, creating more room for the LTM to improve reasoning. Nevertheless, in open-ended generation tasks (Tab. 3), our method already provides a significant accuracy boost for VideoChat2 models.
>
> We are fixing the typos, thank you for pointing them out!

---

### Official Review · Reviewer_iEVz · 2025-03-16

**Overall Recommendation:** 4

**Summary:**

This paper proposes a method called “\infty-VIDEO” to enable large multimodal language models (LLMs), originally designed for short video contexts to process arbitrarily long videos. The approach builds on top of existing “video Q-former” architectures by equipping them with a new continuous-time long-term memory (LTM) mechanism. Specifically, the authors use a “continuous attention” strategy that compresses and consolidates past chunks into a fixed dimension memory representation. Experiments on multiple video QA benchmarks, some of which include minutes-long and even hour-long videos, indicate the method can improve long-video comprehension in a training-free manner.

**Claims And Evidence:**

- Claim: the proposed continuous-time attention can dynamically allocate higher granularity to crucial frames and effectively compress less relevant parts.

  Evidence: qualitative heatmaps of the attention densities show peaks at visually distinctive or narrative-significant frames. Quantitative results indicate improvements over uniform sampling baselines.
- Claim: sticky memories (adaptive sampling guided by prior attention) outperform simple uniform sampling.

  Evidence: in multiple-choice QA tasks, sticky memories yield consistently better accuracy. In open-ended QA (MovieChat-1K), sticky memories often outperform pure uniform sampling.

**Essential References Not Discussed:**

No

**Experimental Designs Or Analyses:**

Yes.  The authors run experiments on NeXT-QA, EgoSchema, Video-MME and MovieChat-1K. Key analyses include accuracy gains from ablating the “sticky” memory vs. “uniform” memory. The paper also includes qualitative visualizations.

**Methods And Evaluation Criteria:**

Yes. The evaluation is sound, including:
- Multiple-Choice Question Answering
- Long-Term Open-Ended Question Answering

**Other Comments Or Suggestions:**

- The authors are recommended to also compare with recent VLM models such as [*1, *2, *3].
- Even the paper focuses on training-free setting, it is also recommended to compare with some trained based long-video understanding paper, such as longvila and Video-xl.
- A breakdown of computational costs would be helpful—for example, how the inference time scales with the number of chunks or basis functions.

[*1] Wang, Peng, et al. "Qwen2-vl: Enhancing vision-language model's perception of the world at any resolution." arXiv preprint arXiv:2409.12191 (2024).

[*2] Zhang, Yuanhan, et al. "Video instruction tuning with synthetic data." arXiv preprint arXiv:2410.02713 (2024).

[*3] Cheng, Zesen, et al. "Videollama 2: Advancing spatial-temporal modeling and audio understanding in video-llms." arXiv preprint arXiv:2406.07476 (2024).

**Other Strengths And Weaknesses:**

**Strengths**
- The paper proposes a training-free method which is novel and barely explored in this field.
- Sticky memory sampling effectively focuses on critical frames, as demonstrated by meaningful attention heatmaps.
- The motivation of the paper is strong and intuitive.

**Weaknesses**
- The paper misses analysis about memory usage and runtime overhead for very large videos. While the paper highlights the advantage of not storing all frames in memory, the scaling behavior and trade-offs could be more explicitly benchmarked.
- The conclusion from Table 4 is not consistent. It is better to test other Q-former video models and give a deeper analysis.

**Questions For Authors:**

- Would segmenting the video by scene change (rather than fixed-length chunks) further boost performance?
- How are the hyperparameters chosen? For example, is the method sensitive to memory contraction factor?
- How sensitive are results to the choice of chunk size vs. number of chunks? Is there an optimal chunk length for typical tasks?

**Relation To Broader Scientific Literature:**

I think the key contribution is the exploration of a training-free approach for long video understanding.

**Theoretical Claims:**

No theoretical claims are made.

---

> ### Author Rebuttal · Authors · 2025-03-31
>
> Thank you for your positive review and suggestions. We are glad that you found our method novel, our evaluation sound, and the motivation of our work strong and intuitive. We address your main concerns below.
>
> > “The paper misses analysis about memory usage and runtime overhead (...) the scaling behavior and trade-offs could be more explicitly benchmarked.”
>
> This is a good suggestion. We agree that it is important to analyze the memory usage and runtime overhead of our method for large videos. Our experiments focus on a fixed-dimensional memory representation that scales independently of video length. To address this concern, we have conducted additional experiments that show the trade-offs in inference time and memory usage as the number of video chunks and basis functions increases, including the impact of the LTM. We will add these results to the appendix.
>
> ### Inference Time vs L
> | **L**|**Video LLaMA (s)**|**Ours, No LTM (s)**|**Ours, LTM (s)**|
> |-|-|-|-|
> |**256**|31.25|31.25|33.10|
> |**512**|36.68|35.55|37.42|
> |**1024**|38.72|40.03|41.41|
> |**2048**|OOM|51.30|55.79|
>
> ### Memory Usage vs L
> |**L**|**Video LLaMA (Gb)**|**Ours, No LTM (Gb)**|**Ours, LTM (Gb)**|
> |-|-|-|-|
> |**256**|18.99|18.99|18.99|
> |**512**|22.67|18.99|19.17|
> |**1024**|30.04|19.46|19.47|
> |**2048**| OOM| 20.05|20.96|
>
> ### Inference Time vs N
> |**L**|**Video LLaMA (s)**|**N**|**Ours (s)**|
> |-|-|-|-|
> |**64**|32.27|**64**|42.94|
> |**128**|33.49|**128**|40.51|
> |**256**|31.25|**256**|41.41|
> |**512**|36.88|**512**|41.68|
> |**1024**|38.72|**1024**|42.18|
>
> ### Memory Usage vs N
> |**L**|**Video LLaMA (Gb)**|**N** |**Ours (Gb)**|
> |-|-|-|-|
> |**64**|16.23|**64**|19.47|
> |**128**|17.15|**128**| 19.47|
> |**256**|18.99|**256**|19.47|
> |**512**|22.67|**512**|19.48|
> |**1024**|30.04|**1024**|20.64|
>
> In the first two tables, for $\infty$-Video LLaMA, we use 256 frames per chunk and set $N=256$, while for the baseline (Video LLaMA), we use the total number of frames without chunking. In the last 2 tables for “Ours” we use 4 chunks of 256 frames. We use the Bohemian Rhapsody movie. OOM denotes an Out-of-Memory error on an A6000 GPU. Results show that our method maintains a constant memory footprint regardless of the number of frames, with only a small increase in inference time. We also observe that the increase in memory usage with the number of basis functions is smaller compared to the increase with the number of frames in the baseline. We hope this alleviates your concern.
>
> > “The conclusion from Table 4 is not consistent. It is better to test other Q-former video models and give a deeper analysis (...) compare with recent VLM models such as [*1, *2, *3].”
>
> Please note that the main goal of our paper is to validate the benefits of integrating a LTM with continuous attention in a fair, training-free setting, for which the Q-former video models are a suitable choice. We focused on an apples-to-apples comparison to isolate the impact of our approach, paving the way for future integration into stronger VLMs, which keep evolving very rapidly. For more details, see our response to Reviewer SJwR (second question).
>
> > “Would segmenting the video by scene change (rather than fixed-length chunks) further boost performance?”
>
> In our early experiments, we tried scene segmentation, but for fast-moving videos, this created small chunks that did not improve performance. Testing different scene-dividing granularities resulted in some chunks containing multiple scenes, which yielded worse results than fixed-length chunks. In any case, we find fixed-length chunks more interesting, since the ability of our method to generate sticky memories without the need for any scene segmentation suggests it is able to identify the most relevant information in the different scenes, bypassing this additional step.
>
> > “How are the hyperparameters chosen? For example, is the method sensitive to memory contraction factor? (...) How sensitive are results to the choice of chunk size vs. number of chunks? Is there an optimal chunk length for typical tasks?”
>
> The number of frames per chunk was determined by balancing GPU memory constraints and alignment with the training setup of the base models. For VideoChat2-based models, we selected 16 frames per chunk, as this both fits within memory limits and also matches the number of frames used during the training of VideoChat2. For Video LLaMA, although the model was originally trained with 32 frames, [1] has demonstrated that increasing this to 256 frames does not degrade performance. We therefore used 8 chunks of 256 frames, which aligns with the total number of frames in MovieChat [1]. Finally, the $\tau$ parameter was chosen to be the same as in the  $\infty$-former paper [2].
>
> [1] Moviechat+: Question-aware sparse memory for long video question answering (Song et al., 2024)
>
> [2] $\infty$-former: infinite memory transformer (Martins et al., 2022)

---

### Decision · Program_Chairs · 2025-05-01

**Decision:**

Accept (poster)

**Comment:**

While the paper has some weaknesses related to newer model comparisons and computational efficiency analysis, it introduces a highly novel and impactful method for long-video understanding without requiring training. The method is well-supported by experimental results, showing solid improvements across various video QA benchmarks. Given the paper's novelty, experimental soundness, and strong theoretical foundation, the recommendation is to accept. Further improvements could include comparing the method with newer models and a more thorough analysis of computational overhead for very large videos.